# Impact of physician' and pharmacy staff supporting activities in usual care on patients' statin adherence

Victor Johan Bernard Huiskes[1]*, Johanna Everdina Vriezekolk[2], Cornelia Helena Maria van den Ende[2], Liset van Dijk[3,4], Bartholomeus Johannes Fredericus van den Bemt[1,5,6]

1 Department of Pharmacy, Sint Maartenskliniek, Nijmegen, The Netherlands, 2 Department of Rheumatology, Sint Maartenskliniek, Nijmegen, The Netherlands, 3 Nivel, Netherlands Institute for Health Services Research, Utrecht, The Netherlands, 4 Department of PharmacoTherapy, -Epidemiology & -Economics (PTEE), Groningen Research Institute of Pharmacy, Faculty of Science and Engineering, University of Groningen, Groningen, The Netherlands, 5 Department of Pharmacy, Radboud University Medical Center, Nijmegen, The Netherlands, 6 Department of Clinical Pharmacy and Toxicology, Maastricht University Medical Center+, Maastricht, The Netherlands

* v.huiskes@maartenskliniek.nl

**Data Availability Statement:** The anonymized data that support the findings of this study are uploaded as Supporting Information file.

## Abstract

### Aims

Little is known about usual care by physicians and pharmacy teams to support adherence to statins and whether the extent of this care is associated with adherence to statins. Objective of the study was to examine the relationship between the extent of adherence supporting activities of healthcare practitioners (HCPs) and patients' adherence to statins.

### Methods

Cross-sectional study in 48 pharmacies and affiliated physicians' practices, between September 3, 2014 and March 20, 2015. Patients visiting the pharmacy with a statin prescription from participating prescribers were invited to participate. Usual care to support adherence was assessed among HCPs with the Quality of Standard Care questionnaire about usual care activities to support adherence. Adherence to statins was assessed among patients with the MARS-5 questionnaire. The association between the extent of HCPs' adherence supporting activities and patients' adherence was examined by means of multilevel regression analysis.

### Results

1,504 patients and 692 HCPs (209 physicians, 118 pharmacists and 365 pharmacy technicians) participated. No association was found between the extent of physicians' adherence supporting activities and patients' adherence to statins. The extent of adherence supporting activities by pharmacy teams in usual care was negatively associated with patients' adherence to statins (B coefficient -0.057 (95%CI: -0.112- -0.002).

**Funding:** The author(s) received no specific funding for this work.

**Competing interests:** There are no conflicts of interest for this manuscript. Although L. van Dijk has received grants for research not related to this study from TEVA and AstraZeneca, this does not alter our adherence to PLOS ONE policies on sharing data and materials.

## Conclusions

This study suggests that there is no positive relationship between the extent of HCPs' adherence supporting activities in usual care and patients' adherence to statins. Other methods than questionnaires (e.g. electronic monitors (to assess adherence) and observations (to assess usual care) should be applied to confirm the results of this study.

## Introduction

Statins are a proven therapy to lower serum cholesterol concentrations, reducing the long-term risk of ischaemic heart disease events by about 60% and stroke by 17% [1]. Despite these therapeutic advantages, medication adherence to statins (defined as the extent to which the patient's medication taking behavior corresponds with the agreed recommendations from the healthcare provider) is suboptimal and varies between 32–77% [2–8].

Non-adherence to statin therapy has a negative impact on treatment outcomes. Patients with poor adherence to statins are at greater risk of cardiovascular events and hospitalization due to cardiovascular disease and cause avoidable high health care costs [9–15]. This makes improving medication adherence to statin therapy a key component of the treatment of hypercholesteremia [9, 16].

Adherence is multifactorial; "Health-system/Health-care team factors", "Social/economic factors", "Condition-related factors", "Therapy-related factors" and "Patient-related factors" have been associated with/implicated in non-adherence [9]. Previous research on interventions to improve adherence to statins mainly focused on "patient-related factors", however these studies yielded small inconsistent results, with a range of effect of these interventions from -3% up to 25% improvement of adherence [17–20]. Therefore, interventions that target other factors that can have impact on adherence might also be required, like relevant factors in the health-system/health-care [9]. Yet, evidence on the impact of health-system/health-care team factors on implementation adherence to statins is scarce. Insight into the association between relevant factors in the health system/health-care team and adherence is warranted.

Earlier studies demonstrated health system factors like continuity of care and complete treatment information are factors that are positively associated with adherence to drug treatment in chronic conditions as well as in statin use [16, 21, 22]. Furthermore, patients who experienced a higher quality of care and/or a higher degree of shared decision making had more knowledge of their illness, were more actively involved in their own treatment, were more confident in their communication with healthcare providers and had higher adherence rates [23, 24]. The aforementioned examples in literature are about the impact of the overall quality of care on adherence, whereas literature about the impact of the quality of care activities employed by individual healthcare practitioners (HCPs) is scarce. Based on the findings about the positive impact of the overall quality of care on adherence, it is also conceivable that quality of care activities, including usual care adherence support activities) of a single HCP, might positively influence patients' medication adherence. Noteworthy, influencing the usual care of one single healthcare provider may affect the adherence of several patients, which makes interventions on HCP level potentially more impactful than interventions on patient level. Currently, no evidence is available about physicians' and pharmacy staff's' usual care to support adherence to statins and how this care affects patients' adherence.

The aim of this study is 1) to describe the nature and extent of adherence supporting activities provided in a usual care setting by physicians, pharmacists and pharmacy technicians; and

2) to examine the relation between the extent of adherence supporting activities of physicians, pharmacists and pharmacy technicians and adherence to statins. We hypothesized that increased HCPs' usual care activities to support statin adherence have a positive impact on patients' implementation adherence to statins.

## Materials and methods

### Study design and setting

This cross-sectional study was conducted between September 3, 2014 and March 20, 2015 in 48 Dutch pharmacies (44 community and 4 outpatient). The EMERGE (ESPACOMP Medication Adherence Reporting Guideline) was used as guidance in reporting this study [25]. The Medical Research Ethics Committee (MREC) of Arnhem- Nijmegen waived official ethical approval (file number: 2021–13158) and assessed the trial (including the verbal consent procedure) as not being subject to the Medical Research Involving Human Subjects Act (WMO).

### Eligibility criteria and selection procedures

A pharmacy technician informed all patients with a prescription for a statin (prescribed by one of the included prescribers) and asked these patients to participate in the study. Patients were included only after verbal informed consent was obtained. Verbal consent was recorded per participant on one registration form per participating pharmacy. If patients did not wish to participate the pharmacy technician noted the reason on the registration form. The verbal informed consent procedure was a pragmatic choice, based on the assumption that completing the questionnaire was an implicit agreement of the patient to participate in the study and to achieve an efficient process at the pharmacy counter. For inclusion criteria, we refer to Huiskes et al. [26].

### Measurements

**Variables and data collection.** Patient data were collected with a hardcopy questionnaire assessing socio-demographic characteristics, medication related information (duration statin use, prescriber) and patient's adherence to statins (see measurement instruments). In this study implementation adherence (defined in the ABC taxonomy of medication adherence) was studied, as current statin users were included [27]. Patients were asked by the dispensing pharmacy technician to fill out the questionnaire in the pharmacy or to return the questionnaire by mail. HCPs' socio-demographic characteristics and HCPs' usual care to support adherence (see measurement instruments) to statins were assessed using a hardcopy questionnaire.

**Outcomes.** An inventory of the nature and extent of adherence supporting activities provided in a usual care setting by physicians, pharmacists and pharmacy technicians and the association between the extent of these HCPs' adherence supporting activities and patients' adherence to statins.

**Measurement instruments.** *Usual care questionnaire*. Usual care to support adherence to statins was assessed with a 47-item questionnaire about usual care activities to support adherence based on the Quality of Standard Care questionnaire as used by de Bruin et al., with modifications to assess usual care in patient care as described by Timmers et al. [28–30]. The list used by Timmers et al. was used in cancer care, so for use in this study the list was adapted to statin therapy by one of the researchers (BvdB). HCPs were asked to score the extent of their care activities they performed to support adherence in the majority of their patients the past six months a) when initiating statin therapy, b) during follow-up visits with patients that

already used statins for a longer period and c) for their patients regardless of whether they used a statin. Four out of the 47 items were qualitative questions and 43 items could be answered with yes or no. Due to the quantitative character of this study the four qualitative questions were not included in the analysis. When the response to a quantitative question was answered with yes, the answer was awarded one point. The questions as presented to the HCPs are shown in Table 2. A sum score was calculated by summing the scores of each question, resulting in a sum score from 0 to 43. Furthermore, in order to create a better understanding of the nature and extent of the usual care activities, usual care activities were grouped to sub scales. Also for these sub scales sum scores were calculated. The sub scales were based on the coding taxonomy provided by the original author: knowledge, awareness, attitude, social influence, self-efficacy, intention formation, action control, facilitation, metascore [29]. A higher sum score indicates a higher quality of the level of usual care.

*Self-reported adherence to statins.* The Medication Adherence Report Scale-5 (MARS-5) consists of five items, mainly addressing intentional non-adherence behavior (4 out of 5 items). The items are rated on a five-point Likert scale (from 1 (always) to 5 (never)), resulting in a summated score of 5–25 [31]. No standard cut-off point to define adherent versus nonadherent medication has been provided by the scale developers and it varies across studies [32]. In this study the MARS-5 cut-off scores of $\geq 23$ and $\geq 24$ to identify adherent patients are both reported, as these are cut-off points that are more often used and because adherence distributions found with the MARS-5 are often highly skewed [33–36].

## Sample size and data analyses

**Data analyses.** Data were analyzed using STATA version 13. Descriptive statistics were provided using mean ($\pm$ SD) or median (p25-p75) values depending on the (non-) parametric distribution of measured variables. P-values $\leq 0.05$ were considered statistically significant.

The association between the extent of HCPs' usual care activities (sum score of the Quality of Standard Care questionnaire) and the adherence (MARS-5 total score) of patients was subjected to multilevel linear regression analyses (see Huiskes et al. [26]). If a healthcare practitioner did not answer one or more items of the usual care questionnaire within the total of usual care activities or within a sub scale, then the respondent was considered as lacking for the calculation of the total sum score or the sum score of that sub scale.

**Sample size.** In this study a convenient sample of 1504 patients was included as described by Huiskes et al. in the methods section [26]. Based on a conservative estimation of one-third non adherent patients in this population, 501 non-adherent patients were expected. As eight independent variables were planned to be included in these multilevel regression analyses, 62 cases per independent variable were available, which means enough power is achieved, even taking into account the variance attributable to the group level (based on an alpha of 0.05, a beta of 0.8).

## Results

### Response rate

A total of 2229 patients visited the HCPs and were asked to participate in the study. Of these patients, 1504 (67.5%) agreed to participate and were included in this study (Table 1).

A total of 734 HCPs were asked to participate in the study, 692 (94.3%) of whom agreed to participate and were included. The response rates to the questionnaires per type of HCP were: 209 out of 225 (92.8%) physicians, 118 out of 119 (99.1%) pharmacists and 365 out of 390 (93.6%) pharmacy technicians. The following prescribers were included: general practitioner (89.5%), general practitioner in training (1.0%), cardiologist (2.9%), internist (1.9%),

Table 1. Baseline characteristics patients and HCPs.

| Parameter | Patient | Physician* | Pharmacists | Pharmacy technicians |
|---|---|---|---|---|
| | n = 1504 | n = 209 | n = 118 | n = 365 |
| Gender (female) [n (%)] | 675 (46.5) | 94 (45) | 71 (60.2) | 353 (98.1) |
| Age (years) [mean (SD)] | 66.8 (9.9) | 49.5 (10.0) | 36.9 (11.0) | 39.7 (11.4) |
| Years of statin use [median (p25 p75)] | 6 (3–10) | n/a | n/a | n/a |
| Years employed [median (p25 p75)] | n/a | 19 (10–26) | 10.3 (10.0) | 16.2 (11.0) |

*General practitioner 89.5%, general practitioner in training 1.0, cardiologist 2.9%, internist 1.9%, neurologist 0.5%, nurse practitioner 1.0%, practice assistant 2.9%, other 0.5%.

neurologist (0.5%), nurse practitioner (1.0%), nurse specialist in primary care (2.9%), others (0.5%). The mean (SD) number of patients per physician and pharmacy were 6,6 (SD± 5.0) and 31.1 (SD±15.0), respectively.

## Patients' adherence to statins

The median (p25-p75) MARS-5 score was 25 (24–25). A total of 1349/1483 (91%) and 1215/1483 (82%) of the patients were adherent to their statins using MARS-5 cut-off scores of $\geq 23$ and $\geq 24$ respectively.

## HCPs' usual care activities to support adherence to statins

HCPs' (physicians, pharmacists and pharmacy technicians) usual care activities to support medication adherence to statins are reported in Table 2. The median usual care activities total scores ranged from 21–23 between the three subgroups (Table 3). The highest median sum scores (as percentage of the maximum sum score) were found on sub scales for attitude and facilitation (for all types of HCPs) and awareness (for physicians). The lowest median sum scores were found on sub scales for action control and social influence (for all HCPs) (Table 3).

The top three most frequently reported usual care activities by physicians were: "Explain what cholesterol is and why raised cholesterol is undesirable", "Explain how often and how long the medication should be used", "Giving feedback about the effect of the statin using laboratory findings". For pharmacy teams this consisted of: "Monitor and/or discuss possible interactions with other drugs", "Discuss the common side effects of the drug ", "Verbal explanation about statins"(Table 2).

## Association between the extent of HCPs' adherence supporting activities and patients' adherence to statins

The extent of adherence supporting activities by pharmacy teams in a usual care setting was negatively associated with patients' adherence to statins (B coefficient -0.057 (95%CI: 0.112–0.002) (Table 4). No association was found between the extent of physicians' adherence supporting activities and patients' adherence to statins (Table 4).

## Discussion

To our knowledge, this is the first study examining the level of usual care by HCPs to support adherence to statins and the impact of the level of usual care on patients' adherence to statins. The results of this study did not confirm the hypothesis that there is a positive relationship between the extent of HCPs' adherence supporting activities in usual care and patients'

**Table 2. Usual care to support adherence to statins as reported by physicians, pharmacists, pharmacy technicians & pharmacy team.**

| | Cat. | % yes phys. (n = 209) | % yes pharm (n = 118) | % yes pharm tech (n = 366) | %yes pharm team (n = 484) |
|---|---|---|---|---|---|
| **Knowledge** | | | | | |
| 1. Explain what cholesterol is and why raised cholesterol is undesirable | S | 96 | 54 | 50 | 51 |
| 2. Discuss what a statin is and the mechanism of action | S | 77 | 92 | 80 | 83 |
| 3.Hand out brochure or written information about statins | S | 12 | 92 | 95 | 94 |
| 4. Discuss drug storage recommendations | S | 3 | 35 | 41 | 39 |
| 5. Explain what to do if a dose is missed | S | 22 | 34 | 44 | 42 |
| 6. Do you ask patients to repeat the received information in their own words regularly, to check whether the information is understood properly? (*Refers to items:1;2;3;4;5;12;15;16;26;31;32;33*) | S | 18 | 19 | 21 | 21 |
| 7.Verbal explanation to the patient | S | 94 | 99 | 98 | 98 |
| 8. Use of illustrative materials (pictures/charts/video) | S | 14 | 4 | 3 | 3 |
| 9. Hand out written information | S | 20 | 94 | 97 | 96 |
| 10. Refer patients to websites | S | 38 | 8 | 7 | 7 |
| 11. Do you ask patients to repeat the received information in their own words regularly, to check whether the information is understood properly? (*Refers to items:13;17;20;21;22;28;29;30*) | S | 19 | 17 | 19 | 18 |
| **Awareness** | | | | | |
| 12. Discuss the consequences of non-adherence | S | 49 | 50 | 48 | 49 |
| 13. Encourage patients to use a 7-day pillbox | S | 47 | 36 | 37 | 37 |
| 14. Giving feedback about the effect of the statin using laboratory findings | F | 95 | 18 | 10 | 12 |
| **Attitude** | | | | | |
| 15. Explain that the patient doesn't notice the effect of the statin but that the effect is evaluated by blood tests to check cholesterol levels | S | 87 | 82 | 79 | 80 |
| 16. Discuss the importance of adherence | S | 77 | 84 | 80 | 81 |
| 17. Encourage patients to be adherent | S | 81 | 77 | 78 | 78 |
| 18. Ask the patient about non-practical problems with taking the medication as prescribed (unwilling to take medication, for example because of misunderstandings about taking medication) | F | 56 | 36 | 35 | 35 |
| 19. In case of non-practical problems, propose solutions to solve these problems (for example discussing the necessity or concerns, referral to nurse practitioner) | F | 67 | 70 | 61 | 63 |
| **Social influence** | | | | | |
| 20. Involve partner and/or relatives in the treatment | S | 30 | 21 | 15 | 16 |
| **Self efficacy** | | | | | |
| 21. Encourage patients to plan ahead (for example for holidays or social activities) | S | 19 | 16 | 26 | 23 |
| 22. Discuss potential barriers regarding adherence and possible ways to overcome them | S | 41 | 42 | 26 | 30 |
| 23. Ask the patient if he/she is taking the medication as prescribed | F | 79 | 67 | 69 | 68 |
| 24. Ask about practical problems with taking medication as prescribed (for example forgetting it or being unable to open the packaging) | F | 29 | 41 | 34 | 36 |
| 25. In case of practical problems, discuss solutions with the patient to reduce these practical problems | F | 56 | 84 | 81 | 82 |
| **Intention formation** | | | | | |
| 26. Explain how often and how long the medication should be used | S | 95 | 93 | 96 | 95 |
| 27. Develop and discuss a written individual dosing schedule | S | 21 | 25 | 17 | 19 |
| 28. Write down patients' dosing schedule (time, name of meds, number of doses) | S | 22 | 41 | 39 | 39 |
| **Action control** | | | | | |
| 29. Identify daily routines (like brushing teeth) and encourage patients to align the taking of medicines with their routines | S | 36 | 35 | 39 | 38 |
| 30. Encourage patients to use alarm devices as a reminder for taking the medication | S | 9 | 16 | 10 | 11 |
| **Facilitation** | | | | | |

*(Continued)*

**Table 2.** (Continued)

| | Cat. | % yes phys. (n = 209) | % yes pharm (n = 118) | % yes pharm tech (n = 366) | %yes pharm team (n = 484) |
|---|---|---|---|---|---|
| 31. Discuss the common side effects of the drug | S | 86 | 97 | 99 | 98 |
| 32. Discuss with the patient how to deal with side-effects | S | 75 | 84 | 93 | 90 |
| 33. Monitor and/or discuss possible interactions with other drugs | S | 63 | 98 | 99 | 99 |
| 34. Discuss the experienced positive effects of the treatment | F | 47 | 38 | 38 | 38 |
| 35. Asking about (perceived) side-effects of the treatment | F | 91 | 82 | 88 | 87 |
| 36. If patients experience side-effects, there is an active contribution to reduce these side-effects (sometimes by providing knowledge or adjusting the treatment) | F | 93 | 92 | 81 | 84 |
| 37. Suggesting a new medication regimen in case patients feel their present regimen is too complex | F | 72 | 84 | 59 | 65 |
| 38. Call the patient after the initiation of the drugs to ask about experiences | G | 9 | 10 | 14 | 13 |
| 39. Give the patient a telephone number and tell who to contact in case of side-effects | G | 23 | 25 | 25 | 25 |
| 40. Give the patient a telephone number and tell who to contact in case of problems with intake/medication adherence | G | 19 | 15 | 21 | 20 |
| 41. Explain patients who to contact in case they would run out of medication | G | 74 | 69 | 84 | 80 |
| **Metascore** | | | | | |
| 42. Intensify the number of follow-up visits in case of (possible) treatment non-adherence | G | 38 | 19 | 12 | 14 |
| 43. Refer patients to another health care provider for (co-)treatment (e.g., in case of side-effects) | G | 35 | 60 | 47 | 50 |

S = when starting statin therapy; F = during follow-up visits; G = in general for their patients regardless of whether they used a statin.

**Table 3. Median scores, interquartile ranges and median scores as percentage of the maximum score.**

| Sub scales* | Min—max | Physicians | | Pharmacists | | Pharmacy technicians | | Pharmacy team** | |
|---|---|---|---|---|---|---|---|---|---|
| | | Median (p25—p75) | Median score as % of max score | Median (p25—p75) | Median score as % of max score | Median (p25—p75) | Median score as % of max score | Median (p25—p75) | Median score as % of max score |
| **Knowledge** | (0–11) | 4 (3–5) | 36 | 5 (4–7) | 45 | 5 (4–7) | 45 | 5 (4–7) | 45 |
| **Awareness** | (0–3) | 2 (1–3) | 67 | 1 (0–2) | 33 | 1 (0–2) | 33 | 1 (0–2) | 33 |
| **Attitude** | (0–5) | 4 (3–5) | 80 | 4 (3–5) | 80 | 4 (3–4) | 80 | 4 (3–4) | 80 |
| **Social influence** | (0–1) | 0 (0–1) | 0 | 0 (0–0) | 0 | 0 (0–0) | 0 | 0 (0–0) | 0 |
| **Self efficacy** | (0–5) | 2 (1–3) | 40 | 2 (1–4) | 40 | 2 (1–3) | 40 | 2 (1–3) | 40 |
| **Intention formation** | (0–3) | 1 (1–2) | 33 | 1 (1–2) | 33 | 1 (1–2) | 33 | 1 (1–2) | 33 |
| **Action control** | (0–2) | 0 (0–1) | 0 | 0 (0–1) | 0 | 0 (0–1) | 0 | 0 (0–1) | 0 |
| **Facilitation** | (0–11) | 7 (5–8) | 64 | 7 (6–8) | 64 | 7 (6–8) | 64 | 7 (6–8) | 64 |
| **Meta-score** | (0–2) | 1 (0–1) | 50 | 1 (0–1) | 50 | 0 (0–1) | 0 | 1 (0–1) | 50 |
| **Sum score*** | (0–43) | 21 (16–26) | 49 | 23 (18–27) | 53 | 21 (17–26) | 49 | 21.5 (18–26) | 49 |

* Respondents were treated as a missing for calculation of the sum score if one or more items were missing. The number of missings was 21%.

** Pharmacy team is the combination of pharmacy technicians and pharmacists.

**Table 4. Multilevel regression analysis for the association between the extent of HCPs' adherence supporting activities and patients' adherence to statins, with controlling for the pharmacy level and physician level.**

| | Patients' MARS-5 adherence scores |
|---|---|
| | B (95% CI) coefficient |
| Adherence supporting activities by physicians | 0.085 (-0.010–0.027) |
| Adherence supporting activities by pharmacy teams | **-0.057 (-0.112- -0.002)** * |

* p $\leq$ 0.05

implementation adherence to statins. The extent of usual care activities hardly differed between physicians, pharmacists and pharmacy technicians. The median sum scores on all sub scales of the Quality of Standard Care questionnaire were comparable for all HCPs, only on awareness physicians scored higher than pharmacy staff.

In this study the level of usual care to support adherence delivered by physicians is comparable and by pharmacists exceeded that reported by Timmers et al. (in patients using oral anticancer drugs) [30]. The latter might be explained by the fact that other HCPs than pharmacists (e.g. nurses) perform these activities (because of differences in setting and type of medication).

In our study, both pharmacists and physicians reported that half of the adherence supporting activities were performed and half were not. When HCPs coordinate their adherence supporting activities, this does not necessarily have to be a problem. This seems to be the case with respect to patient education to improve medication adherence: whereas doctors educate patients about the disease, the effect of the drug and treatment duration, pharmacy staff member tend to focus on adverse events, drug-drug interactions and storage conditions. Although doctors and pharmacy staff members seem to be synergistic with respect to education (sending information), neither doctors nor pharmacy staff members ask the patient about perceived barriers to take the medication as prescribed: patients' knowledge about medication and non-practical barriers and practical barriers taking medication as prescribed are hardly inventoried by both physicians and pharmacy staff.

The extent of usual care of HCPs to support adherence to statins was not positively associated with patients' adherence to statins. This in contrast with two meta-analyses on the quality of usual adherence care and medication adherence in patients infected with Human Immuno-deficiency Virus (HIV) showing that a higher quality of self-reported usual care led to more patients being adherent to their medication [28, 29]. This might be explained by differences in type of medication, and design and setting (cross-sectional inventory of usual care in our study in one country versus retrospective inventory of usual care in usual care arms of trials in several countries). Furthermore, in HIV care often nurses are involved, which requires another role of pharmacists with respect to adherence support. Finally, adherence was measured differently, as in our study the MARS questionnaire was used and in the studies included in the meta-analyses by de Bruijn et al. (2009 and 2010) both self-reported adherence measures and Medication Event Monitoring System (MEMS) devices were used.

The lack of positive impact of usual care of both physicians and pharmacists to support adherence to statins on patients' adherence to statins may be explained by conceptual differences (the extent of unintentional and intentional non-adherence aspects that are incorporated in the questionnaire) between the usual care activity questionnaire and the patient adherence measure (MARS-5). The Quality of Standard Care questionnaire is balanced with respect to the proportion of aspects related to unintentional and intentional non-adherence, whereas the MARS-5 questionnaire used in this study is predominantly focused on intentional non-adherence. Another explanation may be that the overall high MARS-scores might lead to ceiling

effects, which may account for not finding a difference in adherence scores, as described in the strengths and limitations section.

Furthermore, HCPs with a patient population with low adherence rates to statins possibly feel a greater need to perform activities to support adherence to statins and consequently have higher scores on the usual care questionnaire. Alternatively, social desirability bias may have led to an overestimation of the level of usual care reported by pharmacy staff. In that case HCPs provide less activities to support adherence than they say they deliver, tentatively resulting in lower adherence rates and no (or weakly negative) association between the extent of adherence supporting activities and patients' adherence. Participatory observations to assess the actually delivered extent of usual care activities to support adherence could be applied to overcome this.

The current findings should be interpreted in light of the strengths and limitations of our study. One of the strengths of this study concerns the large sample of patients and HCPs, as well as the high response rate, which increases the accuracy of the results. This study was furthermore carried out in a large number of practices across the Netherlands. This last aspect increases the generalizability (with respect to adherence supporting activities of HCPs to stimulate patients' adherence to statins). The fact that the MARS-5 scores of patients using statins in this study were similar to those in another study and that 18% of patients are non-adherent to therapy (similar to the degree of non-adherence in other studies among Dutch patients taking statins), is a prove that a valid sample was included in the study and highlights generalizability [37–39].

However, this study does have its limitations. First of all, self-report questionnaires were the only means used in this study to measure adherence and the level of usual care. Questionnaires of this kind are subjective and therefore sensitive to social desirability bias. It is preferable for that reason to use a combination of methods when measuring adherence (e.g. self-report questionnaires, pill count, refill adherence, medication event monitoring systems and/or biochemical testing) and to observe the HCPs to inventory the level of usual care. If the extent of usual care delivered by a HCP is assessed by observation, it can be decided to observe each HCP once, or to observe all individual patient-provider interactions. Preferably all the individual patient-provider interactions are observed, as the usual care actually provided may depend on a specific patient and/or moment. Seeing that it is likely that adherent patients are more motivated to participate in a study of this kind (confirmed by slightly higher adherence rates in this study than in other studies), inclusion bias may have played a role [3, 8]. The chance that inclusion bias has affected the results, however, is reduced by that fact that the response rate of patients was high (67.5% of the selected patients agreed to participate in the study). Furthermore, we did not collect data on the indication for the use of statins. The extent to which patients perceive future risks and impact on the prognosis due to non-adherent behavior may differ between patient who use the statin for primary or secondary prevention. Therefore this kind of data should be collected and analyzed separately in future research. Another limitation of this study is that due to the cross-sectional design of the study, causality cannot be proven, as this should be investigated in a longitudinal and preferably randomized (to reduce confounding bias) design. Finally, due to a ceiling effect when using the MARS-5 and therefore little explained variance, no difference in adherence scores may be found.

This study provides an overview of usual care activities to support adherence to statins as reported by a large number of physicians, pharmacists and pharmacy technicians employed in a large number of practices in the Netherlands. Furthermore, the results of this study suggest that there is no positive relationship between the extent of HCPs' adherence supporting activities in usual care and patients' adherence to statins. Before trials are performed to improve adherence by intervening on HCPs, first more research with better techniques to objectify the level of usual care to support adherence and the impact on patients' adherence is warranted.

As only questionnaires were used in this study to examine the impact of usual care on adherence, further research in which other methods to measure adherence are used are recommended. Further research could furthermore be supplemented with observing the patient-provider interactions to inventory the level of usual care delivered by HCPs.

## Supporting information

**S1 Data.**
(XLSX)

## Acknowledgments

The authors thank all contributing patients, physicians, pharmacists and pharmacy technicians.

## Author Contributions

**Conceptualization:** Victor Johan Bernard Huiskes, Bartholomeus Johannes Fredericus van den Bemt.

**Data curation:** Victor Johan Bernard Huiskes.

**Formal analysis:** Victor Johan Bernard Huiskes, Johanna Everdina Vriezekolk, Cornelia Helena Maria van den Ende, Bartholomeus Johannes Fredericus van den Bemt.

**Investigation:** Victor Johan Bernard Huiskes, Bartholomeus Johannes Fredericus van den Bemt.

**Methodology:** Victor Johan Bernard Huiskes, Johanna Everdina Vriezekolk, Cornelia Helena Maria van den Ende, Liset van Dijk, Bartholomeus Johannes Fredericus van den Bemt.

**Project administration:** Victor Johan Bernard Huiskes.

**Resources:** Victor Johan Bernard Huiskes.

**Supervision:** Bartholomeus Johannes Fredericus van den Bemt.

**Validation:** Bartholomeus Johannes Fredericus van den Bemt.

**Visualization:** Victor Johan Bernard Huiskes, Bartholomeus Johannes Fredericus van den Bemt.

**Writing – original draft:** Victor Johan Bernard Huiskes, Bartholomeus Johannes Fredericus van den Bemt.

**Writing – review & editing:** Victor Johan Bernard Huiskes, Johanna Everdina Vriezekolk, Cornelia Helena Maria van den Ende, Liset van Dijk, Bartholomeus Johannes Fredericus van den Bemt.

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
