## [Decision Letter · Decision Letter 0]

28 Oct 2021

PONE-D-21-26245Impact of physician’ and pharmacy staff supporting activities in usual care on patients’ statin adherence.PLOS ONE

Dear Dr. Huiskes,

Thank you for submitting your manuscript to PLOS ONE. After careful consideration, we feel that it has merit but does not fully meet PLOS ONE’s publication criteria as it currently stands. Therefore, we invite you to submit a revised version of the manuscript that addresses the points raised during the review process.

We look forward to receiving your revised manuscript.

Kind regards,

Gianluigi Savarese

Academic Editor

PLOS ONE

Journal Requirements:

2. We note that Table 2 may include questionnaire items that may have been previously published. The reproduction of previously published work has implications for the copyright that may apply to these publications. We would be grateful if you could clarify whether you have obtained permission from the original copyright holder to republish these items under a CC BY license. If you have not obtained permission to publish these items please remove them from your manuscript. You may wish to replace the text you have removed with relevant question numbers/ brief descriptions of each item; please be sure to include any relevant references and in-text citations.

3. Please provide additional details regarding participant consent. In the ethics statement in the Methods and online submission information, please ensure that you have specified : 1) whether the ethics committee approved the verbal/oral consent procedure, 2) why written consent could not be obtained, and 3) how verbal/oral consent was recorded. If your study included minors, please state whether you obtained consent from parents or guardians in these cases. If the need for consent was waived by the ethics committee, please include this information.

L. van Dijk has received grants for research not related to this study from TEVA and AstraZeneca 

Please include your updated Competing Interests statement in your cover letter; we will change the online submission form on your behalf."

Reviewers' comments:

Reviewer's Responses to Questions

**Comments to the Author**

1. Is the manuscript technically sound, and do the data support the conclusions?

Reviewer #1: Yes

Reviewer #2: Yes

2. Has the statistical analysis been performed appropriately and rigorously? 

Reviewer #1: I Don't Know

Reviewer #2: Yes

3. Have the authors made all data underlying the findings in their manuscript fully available?

Reviewer #1: Yes

Reviewer #2: Yes

4. Is the manuscript presented in an intelligible fashion and written in standard English?

Reviewer #1: Yes

Reviewer #2: Yes

5. Review Comments to the Author

Reviewer #1: In this interesting analysis the authors assess the association between the level of usual care by HCPs to support adherence to statins and the association between the level of usual care on patients’ adherence to statins. The results did not confirm the authors’ initial hypothesis that there is a positive relationship between the extent of HCPs’ adherence supporting activities and patients’ adherence to statins, also interesting demonstrating a negative association between the level of usual activities among pharmacy teams and patients’ adherence. The study was cross-sectional and was based on questionnaires, i.e. adherence to usual measures was solely self-reported. This is a significant limitation that the authors acknowledge in the respective section of their manuscript.

Considering the journal’s criteria for publication, this study presents the results of original research, which has not been published elsewhere. The methods are described in detail and the analyses are performed to a high technical standard and described in sufficient detail. The conclusions are supported by the data and presented appropriately. The article is presented in an intelligible fashion and is written in standard English.

Based on this I find that the manuscript merits publication after minor revision of the following points:

1) Given the non-randomized design of the study the authors should discuss the highly likelihood that confounding could also explain their unexpected result

2) Page 8: “In this study the MARS-5 cut-off scores of ≥23 and ≥ 24 to identify adherent and non-adherent patients are both reported, as these are cut-off points that are more often used and because adherence”.

Consider omitting the phrase “and non-adherent”

3) Page 9, line 200: reference missing

4) Page 10, Table 1: simplify presentation by abiding to how data are presented based on the statistics section without repeating for each variable

5) Page 17, table 4: β coefficient for adherence supporting activities by pharmacy teams is not within 95%Cis. Please amend

Reviewer #2: The research paper “Impact of physician’ and pharmacy staff supporting activities in usual care on patients’ statin adherence” describes the results of a cross-sectional study that assessed the association between the extent of health care providers’ (HCP) activities aimed at improving adherence to statin therapy and patients’ self-reported adherence in a sample of 1,504 patients and 692 HCPs. Somewhat suprisingly, the results indicate that there is no positive association between HCP efforts to promote adherence and the actual adherence rate reported by the patients. The study is methodologically sound, and used validated measurement instruments and appropriate statistical analyses.

Major comments

1. When investigating the adherence to medications, it is praticularly important to provide details about the indications for their use, which may have an impact on patient’s adherence, because their behaviour may be influenced by the extent to which they perceive their future risks and the impact of treatment on the prognosis. This information is lacking from the manuscript. Could the authors provide description of the patient population and the proportion of patients with statins prescribed in primary vs. secondary prevention?

2. Along the same lines, were there any differences between patients receiving statins for primary prevention vs. secondary prevention?

3. Were there any individual aspects of HCP activities (e.g. knowledge, awareness, attitude etc) that were positively associated with patients’ adherence as those may be the activities further promoted in order to positively influence patients’ behaviour?

Minor comments

Spell out abbreviations at first appearance.

6. PLOS authors have the option to publish the peer review history of their article (what does this mean?). If published, this will include your full peer review and any attached files.

Reviewer #1: No

Reviewer #2: No

---

## [Author Response · Author response to Decision Letter 0]

26 Jan 2022

We would like to thank the editor and the reviewers for their careful and thorough reading of this manuscript and for the constructive suggestions that helped us to further improve it. This document provides a point by point response to each comment and highlights which adjustments were made.

Journal Requirements:

Reply: thank you, we have adjusted the manuscript and file naming according to PLOS ONE’s style requirements.

2. We note that Table 2 may include questionnaire items that may have been previously published. The reproduction of previously published work has implications for the copyright that may apply to these publications. We would be grateful if you could clarify whether you have obtained permission from the original copyright holder to republish these items under a CC BY license. If you have not obtained permission to publish these items please remove them from your manuscript. You may wish to replace the text you have removed with relevant question numbers/ brief descriptions of each item; please be sure to include any relevant references and in-text citations. 

Reply: Thank you for this remark. We have obtained permission from the original author to republish the original questionnaire items under a CC BY license. We have added the name of the original author in the methods section as follows:

Usual care to support adherence to statins was assessed with a 47-item questionnaire about usual care activities to support adherence based on the Quality of Standard Care questionnaire as used by de Bruin et al., with modifications to assess usual care in patient care as described by Timmers et al. (28-30). The latter was used in cancer care, so for use in this study the list was adapted to statin therapy by one of the researchers (BvdB). 

3. Please provide additional details regarding participant consent. In the ethics statement in the Methods and online submission information, please ensure that you have specified : 1) whether the ethics committee approved the verbal/oral consent procedure, 2) why written consent could not be obtained, and 3) how verbal/oral consent was recorded. If your study included minors, please state whether you obtained consent from parents or guardians in these cases. If the need for consent was waived by the ethics committee, please include this information.

Reply: Thank you for this comment. We have added to the methods section that 1) the ethics committee approved the verbal consent procedure, 2) why verbal consent was obtained and 3) how verbal consent was recorded. We did not include minors in this study and the need for consent was not waived by the ethics committee.

We have added the following to the methods section with respect to 1) the ethics committee approved the verbal consent procedure:

The Medical Research Ethics Committee (MREC) of Arnhem- Nijmegen waived official ethical approval (file number: 2021-13158) and assessed the trial (including the verbal consent procedure) as not being subject to the Medical Research Involving Human Subjects Act (WMO).

And with respect to 2) why verbal consent was obtained and 3) how verbal consent was recorded:

A pharmacy technician informed all patients with a prescription for a statin (prescribed by one of the included prescribers) about the study and asked to these patients to participate. Patients were included only after verbal informed consent was obtained. Verbal consent was recorded per participant on one registration form per participating pharmacy. If patients did not wish to participate the pharmacy technician noted the reason on the registration form. The verbal informed consent procedure was a pragmatic choice, based on the assumption that completing the questionnaire was an implicit agreement of the patient to participate in the study and to achieve an efficient process at the pharmacy counter.

L. van Dijk has received grants for research not related to this study from TEVA and AstraZeneca 

Please include your updated Competing Interests statement in your cover letter; we will change the online submission form on your behalf."

Reply: Thank you. We have added the statement to the competing interests section. We also updated our competing interests statement in the cover letter.

Reply: Thank you for this comment. We have uploaded the anonymized data set as Supporting Information file. We also adjusted the Data Availability Statement in the manuscript.

Reviewers' comments:

Reviewer's Responses to Questions

Comments to the Author

1. Is the manuscript technically sound, and do the data support the conclusions?

Reviewer #1: Yes

Reviewer #2: Yes

2. Has the statistical analysis been performed appropriately and rigorously?

Reviewer #1: I Don't Know

Reviewer #2: Yes

3. Have the authors made all data underlying the findings in their manuscript fully available?

Reviewer #1: Yes

Reviewer #2: Yes

4. Is the manuscript presented in an intelligible fashion and written in standard English?

Reviewer #1: Yes

Reviewer #2: Yes

5. Review Comments to the Author

Reviewer #1: In this interesting analysis the authors assess the association between the level of usual care by HCPs to support adherence to statins and the association between the level of usual care on patients’ adherence to statins. The results did not confirm the authors’ initial hypothesis that there is a positive relationship between the extent of HCPs’ adherence supporting activities and patients’ adherence to statins, also interesting demonstrating a negative association between the level of usual activities among pharmacy teams and patients’ adherence. The study was cross-sectional and was based on questionnaires, i.e. adherence to usual measures was solely self-reported. This is a significant limitation that the authors acknowledge in the respective section of their manuscript.

Considering the journal’s criteria for publication, this study presents the results of original research, which has not been published elsewhere. The methods are described in detail and the analyses are performed to a high technical standard and described in sufficient detail. The conclusions are supported by the data and presented appropriately. The article is presented in an intelligible fashion and is written in standard English.

Based on this I find that the manuscript merits publication after minor revision of the following points:

1) Given the non-randomized design of the study the authors should discuss the highly likelihood that confounding could also explain their unexpected result

Reply: Thank you for this suggestion. We have added this to the discussion of the manuscript as follows:

Another limitation of this study is that due to the cross-sectional design of the study, causality cannot be proven, as this should be investigated in a longitudinal and preferably randomized (to reduce confounding bias) design. 

2) Page 8: “In this study the MARS-5 cut-off scores of ≥23 and ≥ 24 to identify adherent and non-adherent patients are both reported, as these are cut-off points that are more often used and because adherence”.

Consider omitting the phrase “and non-adherent”

Reply: Thank you, we have removed the phrase “and non-adherent”.

3) Page 9, line 200: reference missing

Reply: Thank you, we have added the reference.

4) Page 10, Table 1: simplify presentation by abiding to how data are presented based on the statistics section without repeating for each variable

Reply: Thank you for this suggestion, we have adjusted Table 1 based on your input.

5) Page 17, table 4: β coefficient for adherence supporting activities by pharmacy teams is not within 95%Cis. Please amend

Reply: Thank you very much, we have added the minus signs that we unfortunately forgot.

Reviewer #2: The research paper “Impact of physician’ and pharmacy staff supporting activities in usual care on patients’ statin adherence” describes the results of a cross-sectional study that assessed the association between the extent of health care providers’ (HCP) activities aimed at improving adherence to statin therapy and patients’ self-reported adherence in a sample of 1,504 patients and 692 HCPs. Somewhat suprisingly, the results indicate that there is no positive association between HCP efforts to promote adherence and the actual adherence rate reported by the patients. The study is methodologically sound, and used validated measurement instruments and appropriate statistical analyses.

Major comments

1. When investigating the adherence to medications, it is praticularly important to provide details about the indications for their use, which may have an impact on patient’s adherence, because their behaviour may be influenced by the extent to which they perceive their future risks and the impact of treatment on the prognosis. This information is lacking from the manuscript. Could the authors provide description of the patient population and the proportion of patients with statins prescribed in primary vs. secondary prevention?

Reply: Thank you very much for this excellent comment. We fully agree that the medication taking behaviour of patients may be influenced by, besides practical barriers, the extent to which patients perceive their future risks and the impact of treatment on the prognosis (perceptual barriers). We also agree that these perceptual barriers in this population may be different for patients using their cholesterol-lowering medication for primary or secondary prevention and that consequently the extent of usual care to support statin adherence indeed may have different impact on the adherence of patients using statins for primary or secondary prevention. However, we did not collect this data, so unfortunately we will not be able to report this data. Future research has to be supplemented with this data. Therefore, based on your comments, we have added this point to the limitations section of the discussion:

Furthermore, we did not collect data on the indication for the use of statins. The extent to which patients perceive future risks and impact on the prognosis due to non-adherent behaviour may differ between patient who use the statin for primary or secondary prevention. Therefore this kind of data should be collected and analyzed separately in future research.

2. Along the same lines, were there any differences between patients receiving statins for primary prevention vs. secondary prevention? 

Reply: Thank you, please see our answer to your first question.

3. Were there any individual aspects of HCP activities (e.g. knowledge, awareness, attitude etc) that were positively associated with patients’ adherence as those may be the activities further promoted in order to positively influence patients’ behaviour?

Reply: Thank you for this interesting question. Indeed we have performed an explorative analysis on the association of individual usual care items and patients’ statin adherence. See the results of this analysis in the table below. However, to avoid multiple testing, we did not include this analysis in our analysis plan and results.

 Physicians Pharm Pharm Tech Pharm team

Domain Category Nr Question MARS B (95% CI) coefficient 

Knowledge Start 1 Explain what cholesterol is and why raised cholesterol is undesirable -.462 (-.917 - -.007) -.403 (-.792 - -.014)

 2 Discuss what a statin is and the mechanism of action 

 3 Hand out brochure or written information about statins 

 4 Discuss drug storage recommendations -.497 (-.895 - -.100)

 5 Explain what to do if a dose is missed -.474 (-.920 - -.028) -.598 (-1.002 - -.194)

 6 Do you ask patients to repeat the received information in their own words regularly, to check whether the information is understood properly? (Refers to items:1;2;3;4;5;12;15;16;26;31;32;33) 

 7 Verbal explanation to the patient 

 8 Use of illustrative materials (pictures/charts/video) 

 9 Hand out written information 

 10 Refer patients to websites -1.409 (-2.143 - -.675) 

 11 Do you ask patients to repeat the received information in their own words regularly, to check whether the information is understood properly? (Refers to items:13;17;20;21;22;28;29;30) 

Awareness Start 12 Discuss the consequences of non-adherence 

 13 Encourage patients to use a 7-day pillbox 

 Follow-up 14 Giving feedback about the effect of the statin using laboratory findings .889 (.365 – 1.412) -2.627 (-4.967 - -.288) -1.314 (-2.483 - -.144)

Attitude Start 15 Explain that the patient doesn't notice the effect of the statin but that the effect is evaluated by blood tests to check cholesterol levels 

 16 Discuss the importance of adherence 

 17 Encourage patients to be adherent 

 Follow-up 18 Ask the patient about non-practical problems with taking the medication as prescribed (unwilling to take medication, for example because of misunderstandings about taking medication) -.533 (-.957 - -.109) -.507 (-.933 - -.080)

 19 In case of non-practical problems, propose solutions to solve these problems (for example discussing the necessity or concerns, referral to nurse practitioner) 

Social influence Start 20 Involve partner and/or relatives in the treatment 

Self-efficacy Start 21 Encourage patients to plan ahead (for example for holidays or social activities) -.864 (-1.491 - -.238) -.876 (-1.497 - -.255)

 22 Discuss potential barriers regarding adherence and possible ways to overcome them 

 Follow-up 23 Ask the patient if he/she is taking the medication as prescribed 

 24 Ask about practical problems with taking medication as prescribed (for example forgetting it or being unable to open the packaging) -.499 (-.959 - -.039)

 25 In case of practical problems, discuss solutions with the patient to reduce these practical problems .242 (.018 - .467) 

Intention formation Start 26 Explain how often and how long the medication should be used .520 (.064 - .977) 

 27 Develop and discuss a written individual dosing schedule -1.216 (-2.216 - -.217

 28 Write down patients' dosing schedule (time, name of meds, number of doses) -.610 (-.978 - -.242) -.622 (-1.014 - -.230)

Action control Start 29 Identify daily routines (like brushing teeth) and encourage patients to align the taking of medicines with their routines 

 30 Encourage patients to use alarm devices as a reminder for taking the medication 

Facilitation Start 31 Discuss the common side effects of the drug 

 32 Discuss with the patient how to deal with side-effects 

 33 Monitor and/or discuss possible interactions with other drugs 

 Follow-up 34 Discuss the experienced positive effects of the treatment -.218 (-.434 - -.002) -.508 (-.926 - -.091) -.462 (-.890 - -.035)

 35 Asking about (perceived) side-effects of the treatment .433 (.068 - .799) 

 36 If patients experience side-effects, there is an active contribution to reduce these side-effects (sometimes by providing knowledge or adjusting the treatment) .650 (.237 – 1.064) 

 37 Suggesting a new medication regimen in case patients feel their present regimen is too complex 

 General 38 Call the patient after the initiation of the drugs to ask about experiences -1.205 (-2.344 - -.066) 

 39 Give the patient a telephone number and tell who to contact in case of side-effects -.710 (-1.234 - -.186) -.807 (-1.447 - -.167)

 40 Give the patient a telephone number and tell who to contact in case of problems with intake/medication adherence -1.111 (-1.774 - -.449) 

 41 Explain patients who to contact in case they would run out of medication 

Meta-score General 42 Intensify the number of follow-up visits in case of (possible) treatment non-adherence 

 43 Refer patients to another health care provider for (co-)treatment (e.g., in case of side-effects) 

Minor comments

Spell out abbreviations at first appearance.

Reply: Thank you, we have spelled out abbreviations at first appearance now.

6. PLOS authors have the option to publish the peer review history of their article (what does this mean?). If published, this will include your full peer review and any attached files.

Do you want your identity to be public for this peer review? For information about this choice, including consent withdrawal, please see our Privacy Policy.

Reviewer #1: No

Reviewer #2: No

---

## [Decision Letter · Decision Letter 1]

14 Feb 2022

Impact of physician’ and pharmacy staff supporting activities in usual care on patients’ statin adherence.

PONE-D-21-26245R1

Dear Dr. Huiskes,

We’re pleased to inform you that your manuscript has been judged scientifically suitable for publication and will be formally accepted for publication once it meets all outstanding technical requirements.

Kind regards,

Gianluigi Savarese

Academic Editor

PLOS ONE

Additional Editor Comments (optional):

Reviewers' comments:

Reviewer's Responses to Questions

**Comments to the Author**

1. If the authors have adequately addressed your comments raised in a previous round of review and you feel that this manuscript is now acceptable for publication, you may indicate that here to bypass the “Comments to the Author” section, enter your conflict of interest statement in the “Confidential to Editor” section, and submit your "Accept" recommendation.

Reviewer #1: All comments have been addressed

Reviewer #2: All comments have been addressed

2. Is the manuscript technically sound, and do the data support the conclusions?

Reviewer #1: Yes

Reviewer #2: Yes

3. Has the statistical analysis been performed appropriately and rigorously? 

Reviewer #1: I Don't Know

Reviewer #2: Yes

4. Have the authors made all data underlying the findings in their manuscript fully available?

Reviewer #1: Yes

Reviewer #2: Yes

5. Is the manuscript presented in an intelligible fashion and written in standard English?

Reviewer #1: Yes

Reviewer #2: Yes

6. Review Comments to the Author

Reviewer #1: The authors have addressed all my initial coments in their revised manuscript. I have no further comments.

Reviewer #2: All my comments and suggestions have been adequately addressed. At this point I have no further questions for the authors.

7. PLOS authors have the option to publish the peer review history of their article (what does this mean?). If published, this will include your full peer review and any attached files.

Reviewer #1: No

Reviewer #2: No

---

## [Editor Report · Acceptance letter]

18 Feb 2022

PONE-D-21-26245R1 

Impact of physician’ and pharmacy staff supporting activities in usual care on patients’ statin adherence. 

Dear Dr. Huiskes:

I'm pleased to inform you that your manuscript has been deemed suitable for publication in PLOS ONE. Congratulations! Your manuscript is now with our production department. 

Kind regards, 

on behalf of

Dr. Gianluigi Savarese 

Academic Editor

PLOS ONE